# From Hidden Insights to Better Understanding: Physicians’ Perspectives on Caregivers’ Tacit Knowledge

**DOI:** 10.3390/healthcare14010025

**Published:** 2025-12-22

**Authors:** Sylvia A. Huisman, Kasper Kruithof, Maartje Hoogsteyns, Appolonia M. Nieuwenhuijse, Dick L. Willems, Ilse H. Zaal-Schuller

**Affiliations:** 1Department of Paediatrics, Emma Children’s Hospital, Amsterdam UMC, University of Amsterdam, Meibergdreef 9, 1105 AZ Amsterdam, The Netherlands; 2Zodiak, Prinsenstichting, Kwadijkerpark 8, 1444 JE Purmerend, The Netherlands; 3Department of Ethics, Law & Humanities, Amsterdam UMC, University of Amsterdam, Meibergdreef 9, 1105 AZ Amsterdam, The Netherlands

**Keywords:** tacit knowledge, profound intellectual and multiple disabilities, caregiving, disability health, inclusive healthcare, intellectual disability physicians, health communication, health promotion, knowledge mobilization

## Abstract

**Highlights:**

**What are the main findings?**
People with profound intellectual and multiple disabilities (PIMD) express pain and discomfort through non-verbal signs and depend on caregivers’ readings and interpretations of these signs.Caregivers’ tacit knowledge (TK) refers to a crucial ability for recognizing these subtle signs and irregularities or for sensing when something is wrong, which are typically difficult to communicate with physicians.

**What are the implications of the main findings?**
Caregivers’ TK is essential for physicians, as it provides valuable diagnostic cues and helps start a joint diagnostic process.Medical care in co-production with caregivers depends on trust and partnership to overcome communication barriers and improve the physical and mental well-being of people with PIMD.

**Abstract:**

**Background:** Diagnosing physical and mental health issues in individuals with profound intellectual and multiple disabilities (PIMD) often proves challenging, as these patients are unable to speak for themselves. Caregivers’ tacit knowledge (TK) refers to interpreting non-verbal signs, and is difficult to communicate with physicians. As limited research exists on physicians’ perspectives, we aimed to explore how intellectual disability physicians (ID physicians) perceive and value caregivers’ TK. **Methods:** Semi-structured interviews with ID physicians (n = 10), focusing on their perceptions and experiences with caregivers’ TK in medical care for people with PIMD, were analyzed thematically with an interpretive approach to identify key patterns and insights. **Results:** ID physicians perceived caregivers’ TK as a critical ability to pick up subtle signs and irregularities or as a deep sensing something is wrong, with the latter being more difficult to communicate. They understood the importance of TK for diagnostic cues and of collaborative relationships to explicate TK. **Conclusions:** We describe how integrating caregivers’ TK with medical knowledge relies on trust and partnership. Moreover, we discuss how to overcome communication barriers and to improve medical care in co-production with caregivers in order to enhance the physical and mental well-being of people with PIMD.

## 1. Introduction

‘There is something different about my daughter: I sense something is wrong with her, and I’m so worried, doctor!’ Diagnosing physical and mental health issues in individuals with profound intellectual and multiple disabilities (PIMD) often proves challenging, as these patients are unable to provide an anamnesis. People with PIMD, characterized by an estimated IQ below 25, sensory impairments, and severe motor disabilities, express pleasure, pain, and discomfort through facial expressions, vocal sounds, body language, and subtle behaviours [1,2]. These individuals rely entirely on their caregivers’ expertise to sense, pick-up and interpret their non-verbal expressions and address their needs [3]. This expertise, referred to as tacit knowledge (TK), represents a deeply personalized understanding developed through repeated, intimate interactions within caregiving dyads over time and is difficult to explain or communicate to others [4,5,6].

Given the high prevalence of chronic, recurrent, and complex medical conditions in individuals with PIMD from birth [7,8], caregivers’ TK is crucial in signalling potential medical issues and communicating health concerns to physicians. In the Netherlands, intellectual disability physicians (ID physicians) are trained to deliver medical care tailored to individuals with intellectual disabilities (ID), including those with PIMD [9,10]. However, little research exists on how physicians perceive and integrate caregivers’ TK into medical practice. This study aims to explore ID physicians’ perceptions and valuation of caregivers’ TK in medical care for people with PIMD.

## 2. Methods

### 2.1. Study Design

We conducted qualitative research with semi-structured interviews in which we encouraged physicians to reflect on their experiences with parents and professional caregivers bringing in TK during medical consultations and how they understood such TK in light of diagnosis and treatment.

### 2.2. Researchers’ Positionality

The first and last authors of this study are ID physicians with particular experience with medical care for persons with PIMD. This enabled them to recruit ID physicians with relevant expertise (see Section 2.3) and create rapport during interviews. One of the ID physician researchers had formal training in qualitative research and conducted several qualitative studies, while the other contributed as an early-career qualitative researcher bringing substantial clinical expertise. Both worked within an interdisciplinary research team of experienced qualitative researchers, with whom they engaged in regular reflective and methodological discussions throughout the study.

Their experience as ID physicians made that they were aware of when and how caregivers bring in TK during medical consultations. This helped them to provide clarity, e.g., by giving examples, to respondents, which made the relatively vague phenomenon of TK more concrete for respondents. However, this simultaneously meant that they had to be cautious not to influence the direction of the interview, or that the respondents would not explicitly address important topics as these were deemed self-explanatory. SH and IZ, therefore, refrained from steering the interviews when respondents went off-topic, and encouraged respondents to describe situations in detail. They refrained from giving their own reading of situations, their own presuppositions during interviews, in order to not influence respondents’ answers.

### 2.3. Recruitment

We recruited ID physicians from our own professional network. We sampled purposively, as we had a particular interest in the medical care for persons with PIMD. Therefore, ID physicians were included when they had experience with, and interest in, the medical care for persons with PIMD. ID physicians with a direct and hierarchical working relationship with SH or IZ were not included in the study. We included ID physicians from various (residential and outpatient) clinical settings and from various geographical areas of the Netherlands. Moreover, we selected a mixed sample with various ages and lengths of work experiences, as we believed that these characteristics could be of influence on how physicians understood and valued caregivers’ TK. SH and IZ sent an invitation letter by email with information about the study and a request for an interview to potential respondents. Two weeks after the invitation letter was sent, SH or IZ contacted the ID physicians by phone to answer any remaining questions and to ask them to participate in the study. We invited 12 ID physicians for an interview. Two declined, with time investment as reported reason. Our final sample consisted of ten ID physicians.

### 2.4. Data Collection

Interviews were held by SH and IZ during September–December 2022 and lasted a mean of 60 min (range 50–65 min). They started each interview by briefly explaining the study and the concept of TK as introduced in the invitation letter, which we described as an implicit form of knowing that is difficult to put in words. They used an interview guide with open-ended questions and asked the respondents how they would describe caregivers’ TK, how they valued caregivers’ TK, and how caregivers’ TK was communicated. Throughout the interviews, the respondents were consistently encouraged to provide examples from clinical practice to illustrate their perceptions of and experiences with caregivers’ TK. By asking for practical examples, the interviewers aimed to explore not only how ID physicians perceived caregivers’ TK but also how they valued it based on examples of practical situations. If respondents indicated that they valued or dispraised caregivers’ TK, the interviewers further inquired about specific instances from practice that demonstrated this. This approach allowed for concretisation of the role of caregivers’ TK in ID physicians’ medical practices regarding individuals with PIMD. The interviewed physicians were subsequently asked to elaborate on whether and how caregiver’s TK could be combined with medical knowledge and how that worked out to explore barriers and facilitators that ID physicians experienced in putting caregivers’ TK to use. The full interview guide is available upon request.

### 2.5. Data Analysis

We transcribed the interviews verbatim and analyzed them thematically in MaxQDA [11]. First, SH and IZ familiarized themselves with the data and highlighting segments that were relevant in relation to our research questions. After this, they individually coded the transcripts in open fashion to remain as close as possible to participants’ experiences. SH and IZ discussed their open codes among each other and reached consensus on the main themes and overarching patterns in the data. They presented an overview of the main themes and overarching patterns to the other authors. Minor remarks regarding clarity and coherence of quotations and themes were raised and consequently dealt with by SH. Finally, to further enhance the credibility of the results and validation of our analyses, we reflected upon and discussed our findings and interpretations during different meetings and conferences with parents, professional caregivers, ID physicians, developmental pediatricians, and various other professionals in the care for people with PIMD. Data saturation was achieved as no new insights emerged during later interviews, and consistent patterns were found among the respondents based on consensus within the research team.

### 2.6. Research Ethics

According to Dutch Law on Research Involving Human Subjects, this study did not need approval of a Research Ethics Committee, which was confirmed by the Research Ethics of Amsterdam UMC, location AMC (W21_435 # 21.484). We followed the ethical principles for medical research involving human subjects as laid down in the Declaration of Helsinki [12]. We informed eligible participants orally and by written letter about the research project and about their rights, including the right to withdraw from the study at any moment. We provided the opportunity to ask questions and obtained written or oral consent for publication. All participants’ data were pseudonymized to protect confidentiality.

## 3. Results

### 3.1. Respondents

Ten ID physicians (see Table 1) participated in this study. All respondents (eight females, two men; mean age 48 years, range 31–65 years; mean working experience 12.5 years, range 4–20 years) had a Dutch background. They worked in clinical and outpatient care settings in various geographical areas of the Netherlands. More than half of the participating ID physicians had prior experience in the care for people with ID before starting the ID physicians’ specialist training program, and four ID physicians practiced another medical profession before, like general practice, child and youth healthcare, or anaesthesiology.

### 3.2. ID Physicians’ Perspectives

Following a brief general introduction about patients with PIMD who are unable to communicate verbally and who attend consultations accompanied by parents or professional caregivers who know them intimately, are able to interpret their signals, and speak on their behalf, ID physicians elaborated on how they would characterize this specific form of knowing and interpreting the patient, how this actually works in practice, and what takes place during consultations. Their perceptions and experiences regarding 1. nature, 2. valuation, 3. communication about caregivers’ TK during consultations are described below and presented in a thematic map (Figure 1).

#### 3.2.1. Nature

All ID physicians described how parents and professional caregivers of people with PIMD frequently presented with a form of experiential knowledge to notice something was different than usual or wrong, like pain or discomfort.

*Experiential knowledge. … Well, what you [as a physician] often ask the person, who knows the client well, ‘Is this normal or is this not normal?’* [Physician 5]

*Knowing through experience. … A pretty common example is that parents or caregivers can often tell from a client’s behaviour that they’re in pain or not feeling well.* [Physician 8] 

Several ID physicians argued that this know-how could be seen as a form of communication trough bodily expressions between a person with PIMD and the parent or professional caregiver.

*At its core, it is experiential knowledge and experience in the broadest sense of the word, meaning that through experience, one learns how communication—beyond spoken language—functions… What constitutes communicative signals and what is person-specific, identifying the signals unique to an individual.* [Physician 4]

*I consider that [know-how] more as non-verbal communication—all the communication they [individuals with PIMD] express through their body, but not through words.* [Physician 7]

ID physicians described how this experiential knowledge was difficult to put in words and should be understood as TK.

*And the difficulty is that it cannot be put in words, because it happens on a completely different level.* [Physician 10]

Some described this TK as purely intuitive and fully implicit, and referred to an embodied form of deep sensing, understanding or just knowing. This form of embodied knowing and sensing was seen as particularly hard to articulate and as a quality that parents naturally develop over time to conceive what their child with PIMD may experience by attuning to their child, that they share an intimate and longstanding relation with.

*I think parents need to develop that quality because, especially with this group [individuals with PIMD], you cannot rely on reasoning. You cannot talk with your child; you really have to sense them. So, I think they have… (through years of 24/7 care) naturally developed that skill to a great extent.* [Physician 10]

Others described that caregivers’ TK is the capacity to read, interpret and differentiate subtle signs in fact perceived by observing, hearing, touching the person with PIMD. This form of TK was seen as (partly) explicatable and learned by iterating experience by looking closely and with dedicated interest at the person with PIMD.

*I think I quickly tend to say that it is experience. But also by closely observing someone… and generally being interested in someone.* [Physician 9]

ID physicians connected the ability of caregivers to develop TK to their intimate and longstanding relations with the person with PIMD and to their openness and willingness to understand the person with PIMD. Several ID physicians expressed the importance of affinity, pleasure, and empathy with the person with PIMD.

*Affinity… And also the joy. Affinity is the pleasure, the joy. The ability to connect (emotionally). It is more than just showing empathy… There is also something reciprocal about it. It… reflects just seeing someone as a human being with behaviours…* [Physician 2]

#### 3.2.2. Valuation

All participating ID physicians reported how much they rely on experiential knowledge of parents and professional caregivers who know the person with PIMD well and can be translators of the person’s complaints and needs.

*I really need the person who knows the client well to translate that [behaviour] for me*. [Physician 5]

Several ID physicians emphasized they tend to take parents’ knowledge, for example, in recognizing irregularities or changes, very seriously, because their knowledge has been built on years of experience. Furthermore, these ID physicians reported the significance of caregivers’ TK, that is essential for (early) signalling of medical complaints and that may provide important diagnostic cues.

*I do think that, because of the parents’ history and experience, you have to take it very seriously when they notice changes. But also that—these are often small changes in behaviour that parents observe, which make them think that something might be wrong.* [Physician 8]

In valuing caregivers’ TK most ID physicians also took in account what they knew of the (level of experience of the) caregiver and how the caregivers’ TK is presented during consultations. While ID physicians realized that it is difficult to verbalize caregivers’ TK, they reported that it was particularly helpful and they took caregivers’ TK more seriously when the caregivers were able to substantiate their worries, indicate what exactly was special or unusual about the presentation or behaviour of the person with PIMD and how exactly they had noticed that.

*I also think the justification they provide, the explanation of what they think they see, is important. If there are parents who say, ‘I’m worried’, and they can explain, ‘I see it in her because I notice that…,’ then I think, there is a thought behind it, there is a reason for them to signal this now with the client. … But initially, I will still assess the information based on what I know about the caregiver. And also based on how people convey the information. Can they indicate certain things? For example, ‘He doesn’t look well.’ That could be the case, but what exactly do you see in him? And when there is little concrete information provided, I think, ‘What am I supposed to do with this?’ But if they say, ‘He doesn’t look well because he seems to be sweating at times, or looks anxious, or appears stiff,’ then they can support their feelings with what they observe, and I take that more seriously.* [Physician 9]

In contrast, some ID physicians tended to value caregivers’ TK less highly when presented with strong emotions and personal beliefs that may hamper reliable interpretations of the situation of the person with PIMD.

*But then you notice that they are sometimes so emotionally involved and interpret it as pain or feel that there MUST be something physical going on… And they also translate it a bit from their own perspective, how they themselves would react non-verbally, so they easily interpret it in that way.* [Physician 7]

#### 3.2.3. Communication

Caregivers’ TK is personalized, implicit and typically hard to articulate and communicate. Therefore, it is difficult for the physician to comprehend or to assess its value.

*And you [as a caregiver] need to be able to convey that [TK]. And the complexity is that it cannot be communicated verbally, …. So, how do you communicate about that with each other…* [Physician 10]

Some ID physicians made an extra effort to reach out and to empathize with caregivers, asking questions to better understand and interpret caregivers’ perspectives.

*I always want to try to figure out what they mean or what is going on. In fact, I think I actually work harder in those moments. You start asking more questions to understand exactly what they mean because you either don’t fully grasp it yourself or can’t place it. Then I make an effort to understand or make sense of it.* [Physician 3]

During medical consultations caregivers’ TK is often accompanied with a strong sense something is the problem and presented with serious worries, firm beliefs, and powerful emotions of the caregivers. When caregivers’ TK is the main source of information and this information is difficult to explicate, this could exacerbate caregivers’ worries and emotions like fear and panic and could increase the sense of uncertainty of the physician too. Several ID physicians expressed that they felt the need to acknowledge and address feelings and emotions of the caregivers’ themselves first before exploring the content of the worries and the caregivers’ TK itself.

*There was a lot of panic with the caregivers. Almost excessive… So, you do have to do something with it.* [Physician 9]

Several ID physicians commented that when caregivers’ TK was presented in an emotional way, this would affect how they would engage with the parents and professional caregivers during medical consultations. Some ID physicians described feeling reluctant at times to take the caregiver’s knowledge seriously in such situations but emphasized the importance of showing empathy towards the caregivers and to take their request seriously.

*But the way they present their message has an impact on you. And now I think, ‘When they come again, you push past your own reluctance towards the person presenting the message. But look at the person as a concerned parent and at least take the question seriously.’* [Physician 9]

Physician 3: I always take it seriously… However, I may question their [parents’] concerns and want to ask more questions and, at the very least, ask why they [parents] are concerned. Or what it is that they are worried about. Hopefully, either to immediately alleviate their concerns or to think, ‘This seems plausible, it could be.’ I want to investigate further, either by talking with the professional caregivers to see if they also notice it, or examining the patient to see if I observe it as well, etc.

Several ID physicians pointed out that it is important to explicitly acknowledge caregivers as experts but also to offer leniency that caregivers’ TK is typically difficult to articulate. This attitude reestablished reassurance and increased trust, according to these ID physicians.

*That you acknowledge, ‘I understand that you know your child very well and that you see this clearly. And we definitely need to find out what is going on, because we know something is wrong.’* [Physician 7]

*If you just have an open conversation with them [caregivers] about it [TK], then everything is fine. You can also restore their trust this way…* [Physician 5]

Some ID physicians had learned that storytelling can be helpful to explicate caregivers’ TK.

*So, listening to life stories and what [subtle and idiosyncratic signs] someone [person with PIMD] shows, I have really taken that into this work. And it has helped me a lot to apply that from the very beginning.* [Physician 10]

Most ID physicians instigated demonstrations by the caregivers and joint observations to explore subtle signs and to help to explicate together what could be the problem.

*Especially [in medical consultations] with parents, I often join them with their child and say, ‘Show me and tell me what you see, and I’ll tell you what I see, and then we’ll figure out what to do about it.’* [Physician 9]

According to ID physicians, caregivers’ TK meets with medical knowledge during consultations. Just recognizing the imperfections of both physicians’ medical knowledge and caregivers’ experiential knowledge, several ID physicians called for adding up these different but complementary types of knowledge to improve diagnosing and medical care in co-production.

*[Medical] Theoretical knowledge is also necessary. And this [caregiver’s TK] is experiential knowledge, experience or, so—… So, you need that knowledge, but whether you get it from books, then we call it knowledge from scientific research. And it’s great that it exists, as it is a very large and important part. But if you stop there, then you miss…, that experiential knowledge, which cannot be captured in words.* [Physician 10]

*I think it [caregivers’ TK] is not instead of… It is an additional tool, I believe, to establish a diagnosis.* [Physician 8]

Some ID physicians warned that a conservative attitude of parents and caregivers regarding their TK hampers the coproduction and integration of caregivers’ TK with their medical knowledge.

*Most of the time, it [caregivers’ TK] is very valuable, unless people get stuck in it … When they are not open to other possibilities of what it could be. So, if caregivers or parents have already completely decided what it must be, and you can’t prove it or you think, ‘I really don’t believe that at all,’ then it becomes difficult if you can’t have a dialogue about it.* [Physician 7]

ID physicians described how parents’ willingness and capability to reflect upon their interpretations, and to leave room for discussion and further evaluation, was important for such coproduction.

*So, parents who naturally weigh the pros and cons and aren’t immediately very definitive—there is more room for investigation with them, I notice in practice.* [Physician 6]

ID physicians stressed how communication about caregivers’ TK benefits from continuity, longstanding relationships and personal interactions between caregivers and physicians, and the buildup of shared understanding and mutual trust among them.

*And that is, of course, one of the things in our care [for people with PIMD]… continuity. It’s so valuable when you— and parents also really appreciate it—have the same doctor over time, because eventually, this doctor will learn to notice the differences [subtle signs and irregularities].*
[Physician 2]

## 4. Discussion

### 4.1. Summary of Findings

This study aimed to explore how ID physicians perceive the nature of caregivers’ TK in the medical care for individuals with PIMD, how they value such knowledge, and how they communicate about it with caregivers.

ID physicians recognized caregivers’ TK as a form of experiential knowledge that is hard to explicate and developed through longstanding relationships and close interactions with the person with PIMD. They referred to this form of knowledge or know-how as the capacity to pick up subtle signs and irregularities, to interpret idiosyncratic reactions, and to differentiate between meanings in the behaviour of individuals with PIMD. Some described caregivers’ TK as an embodied form of deep sensing, understanding or just knowing. Others noted that subtle signs were perceived through observing, hearing, and touching the person with PIMD, with some describing this as a form of communication through bodily expressions. The first form of embodied knowing and deep sensing was regarded especially difficult to articulate by caregivers, whereas the second form was (partially) explicatable.

The ID physicians emphasized that they took caregivers’ TK very seriously, yet they assessed its quality by considering the caregivers’ level of experience and expertise. They found caregivers’ TK to be more effective when it included indications or “subtitles” for idiosyncratic reactions and less effective when it was presented with firm personal beliefs and powerful emotions, like fear and panic.

ID physicians highlighted the importance of caregivers’ TK in providing diagnostic cues and to help explicate caregivers’ TK by further inquiries, joint observations, and measurements to uncover potential issues or diagnose medical problems. Caregivers’ TK and medical knowledge were regarded as complementary. Integration of both knowledge types seems to rely on collaborative relationships based on mutual trust and open communication between caregivers and physicians.

### 4.2. Strengths and Limitations

To the best of our knowledge this is the first study aiming to explore ID physicians’ perceptions and experiences regarding the nature, valuation, and communication of caregivers’ TK in the medical care for people with PIMD. We gathered rich data on the topics discussed in this paper. The semi-structured interviews revealed a comprehensive picture of the physicians’ perspectives on caregivers’ TK, and a differentiated understanding of effective use of caregivers’ TK during medical consultations.

However, the involvement of ID physicians SH and IZ in conducting the interviews may have introduced potential researcher bias. To mitigate this, they briefly introduced the concept of TK in the interviews, allowing respondents to characterize and elaborate on caregivers’ TK based on their own experiences. They were careful not to steer the interviews, instead encouraging respondents to describe situations in detail. Additionally, the analytic process was conducted and critically discussed within an interdisciplinary research group with ethics and social sciences scholars with expertise in qualitative research and experience with people with PIMD. This interdisciplinary dialogue strengthened the robustness of the analysis. While our study did not employ methodological triangulation, the findings were cross-referenced with empirical and theoretical work from our previous studies on caregivers’ TK in this broader research project.

For our study, we purposively recruited ID physicians from our own professional network, as we were specifically interested in the medical care for persons with PIMD. This approach may have introduced social desirability bias, as respondents could have felt a particular affinity for the interviewers and the topic, potentially leading to more positive evaluations of caregivers’ TK. However, caregivers’ TK is likely to play a crucial role in the daily practice of all ID physicians, and the experiences shared by these ten physicians likely hold meaning for the broader professional group.

On the other hand, recruiting ID physicians could be understood as a strength because ID physicians are specialized in providing medical care for individuals with ID, including PIMD, and are trained in communication with these individuals and their caregivers [13]. By studying the perceptions and experiences of ID physicians, we were able to evaluate our hypothesis that caregivers’ TK plays a pivotal role in medical practice. However, our sample of ID physicians could simultaneously be understood as a limitation as this specialism does not exist in other countries. As ID physicians may be more inclined to accept caregivers’ TK than other types of physicians, this implies some caution with interpreting the international generalizability of our findings. However, during meetings with other physicians, such as developmental pediatricians, and various other professionals in the care for people with PIMD, they supported our findings, interpretations and their implications for daily practice.

Our modest sample size could be regarded as a limitation as well. However, we reached data saturation on the topics discussed. Moreover, the findings of this study are in line with previous findings [4,14,15].

### 4.3. The Emergence of Caregivers’ TK in PIMD Studies

The concept of TK was introduced by Polanyi in 1966, who argued that there are instances where we know how to do something without being able to explain why it works [16]. Smith defines TK as “practical, action-oriented knowledge or ‘know-how,’ based on practice, acquired by personal experience, seldom expressed openly, and often resembling intuition.” [17]. According to Schön [18], professionals possess “knowing-in-action” (implicit and situated knowledge) and also engage in reflection-in-action and reflection-on-action to evaluate and refine this knowledge. While TK is a relatively new concept in research on the care and support for persons with PIMD, this form of knowledge has been explored in various healthcare fields, including medical and nursing studies, and generally recognized as a necessary element of being an expert [19,20]. Our study is part of a series of pioneering studies on caregivers’ TK in the care for persons with PIMD that aims to explore its nature and operationalize this type of knowledge in daily practice. In an interpretative literature study, we reported that caregivers’ TK is built in caregiving dyads between caregivers and care-recipients with PIMD who become sensitive and responsive to each other’s cues and together craft care routines, which is necessary to learn to recognize irregularities in daily routines and potential medical issues [4]. Furthermore, based on our findings from semi-structured interviews with parents, we described parents’ TK as the capacity to read subtle signs, or to sense the situation of their child, that is crucial to ensure their children’s (medical) needs are met [14]. In addition, we demonstrated with a qualitative inductive, thematic analysis of audio-taped consultations that parents’ TK, based on sensing and interpreting of behavioural changes, is conveyed to physicians during medical consultations and incorporated in medical decision-making. This highlights the significance of parents’ TK in medical care and underscores the importance of physicians validating parents’ TK while relating it to their medical knowledge [15]. To improve medical care for people with PIMD, we advocate for fostering caregivers’ TK and integrating caregivers’ TK with medical knowledge. Recent insights in pediatric practice further support that parents or caregivers are well positioned to detect early, subtle signs. Caregiver concern about clinical deterioration is strongly linked to critical illness and ICU admission in pediatric patients, prompting proactive assessment and integration of caregiver concern in response systems [21].

### 4.4. Caregivers as Experts

Our findings show that ID physicians view caregivers of individuals with PIMD as experts with a crucial role in detecting when something is wrong and conveying these signals to physicians. This perspective aligns with previous studies that recognize caregivers as experts by experience in interpreting idiosyncratic cues, distinguishing between signals such as pain and discomfort versus pleasure and happiness and helping to identify potential medical issues [5,22].

ID physicians in our study reported evaluating the quality of caregivers’ TK during medical consultations by considering the caregivers’ level of experience and expertise. They regarded parents as experts due to their lifelong involvement and professional caregivers as experts because of their close interactions during daily care routines. Hoogsteyns et al. argued that the process of TK development and conveyance follows phases: 1. trial and error; 2. becoming responsive; 3. refining care and becoming an expert [4]. Through this process, caregivers gradually become experts, reaching a point where they guide others, including physicians, in interpreting cues, while their role as advocates becomes more pronounced [5,23,24]. Due to staffing shortages and frequent personnel changes, professional caregivers for individuals with PIMD may not always reach the third phase of becoming an expert. Conversely, parents of individuals with PIMD living in care facilities may also face challenges in picking up and accurately interpreting signals, as they are no longer involved in every daily routine. It is both important and reasonable that ID physicians recognize caregivers as experts and caregivers’ TK as a valuable form of expertise. At the same time, physicians should be mindful of the factors influencing the development of caregivers’ TK and carefully considering the caregivers’ level of experience and expertise.

### 4.5. Trust and Partnership

ID physicians regarded integration of caregivers’ TK and medical knowledge as relying on collaborative relationships based on mutual trust and open communication. In such collaborative relationships, in which caregivers are invited by physicians to share and discuss their interpretations of the situation of the person with PIMD and are encouraged to relate these interpretations to previous experiences and medical measurements, caregivers’ TK can be strengthened, and its conservative element can be diminished [4,25,26]. Parents described how affirmation of physicians who had trusted their caregiver’s TK helped them to become more confident about their TK [14]. In addition, caregivers’ TK could be used to complement and deepen medical knowledge [5]. Moreover, combining perspectives would decrease physicians’ reluctancy and increase the acceptability of caregivers’ TK by physicians and result in increased coproduction between caregivers’ TK and medical knowledge [22,27]. Therefore, it is indispensable that caregivers and physicians depend on a process of trial and error guided by mutual trust and careful suspicion, repeated reflection, and revaluation, allowing cues to be thoroughly investigated and validated for accurate diagnosis and optimal medical care. Such coproduction could contribute to increasingly accurate interpretations of the situation and needs of individuals with PIMD and consequently result in increasingly tailored ways of providing medical care for them. This study and other studies of our research team [14,15] illustrated examples of this, namely parents and professional caregivers who used their caregivers’ TK to signal, at an early stage, when the person with PIMD needed medical attention.

### 4.6. Effective Communication

Communication about caregivers’ TK between caregiver and physician seems pivotal, as we demonstrated the important role of caregivers’ TK in the (medical) care for people with PIMD in previous studies of our project [4,14,15]. Hoogsteyns et al. [4] and Kruithof et al. [14] described how caregivers’ TK can be built and transferred within the context of a caregiving dyad, while Zaal-Schuller et al. [15] and this present study focused on how this TK can be presented and shared in the context of medical consultations. Figure 2 illustrates thematically a continuum how caregivers’ TK can be built and transferred by iteration in a caregiving dyad and how it can be presented and shared by repeated evaluation during medical consultation in the context of parallel partnerships.

Our findings on ID physicians’ perspectives on caregivers’ TK imply several potential barriers and subsequent facilitators in communication about caregivers’ TK between caregivers and physicians.

#### 4.6.1. Barriers

Firstly, the apparent contrast between the nature of caregivers’ TK and medical knowledge may create initial tension and misunderstanding. Secondly, caregivers’ TK is typically difficult to articulate, which in turn makes it challenging for the receiving physician to comprehend. Thirdly, caregivers’ TK is frequently accompanied by strong emotions such as fear and panic that need to be acknowledged and addressed to make room for caregivers’ TK content. Fourthly, caregivers’ TK sometimes seems conservative, which may hamper mutual reflections and accurate interpretations of the situation and medical needs of the individual with PIMD.

#### 4.6.2. Facilitators

To overcome these potential barriers, ID physicians considered it is helpful if physicians explicitly recognize caregivers’ TK as complementary to medical expertise. They emphasized to take caregivers’ concerns and suspicions seriously and to acknowledge their TK as expertise, and to help to explicate caregivers’ TK as it is inherently challenging to articulate. Both physicians and caregivers should make mutual efforts to explicate caregivers’ TK as clearly as possible, allowing room for revaluations and alternative interpretations [4,25]. Furthermore, physicians should be sensitive and responsive to caregivers’ feelings of uncertainty to reestablish reassurance and to increase trust. Physicians should pay special time and attention to powerful emotions like fear and panic, and reluctance, creating a supportive environment for an open dialogue. ID physicians reported that tools like storytelling, demonstrations and joint observations can help explicating caregivers’ TK in synch with the literature [5,14]. Medical measurements can be used to support caregivers’ TK.

Barriers and subsequent facilitators influencing effective use of caregivers’ TK during medical consultations are thematically illustrated in Figure 3.

### 4.7. Knowledge Mobilization

Growing attention to the relevance of caregivers’ TK urges dissemination and implementation of this knowledge in clinical practice and policy, referred to as knowledge mobilization (KM) [28]. Partnership is a key component of effective and impactful KM and involves meaningfully engaging caregivers and physicians as partners. These partners co-produce knowledge and provide context on how to use this knowledge in practice [28]. Building on Eikelboom et al. [29], advancing toward partnership and person-centered care, necessitates recognizing caregivers’ TK and their role as experts by experience (EBE) in the education of medical students and physicians. Involving EBEs in medical education programs enhances medical students’ and physicians’ development of knowledge, skills, and attitudes, promoting person-centeredness while making the educational experience more engaging, meaningful, and transformative. For caregivers of persons with PIMD as EBEs, participation offers opportunities to gain insights into their own experiences, strengthen relationships with physicians, connect with peers, and feel valued and empowered [29,30,31]. Communication and partnership can be enhanced through educational activities such as conversations, storytelling, demonstrations, joint observations, speed dating sessions with EBE’s, and home visits, which provide opportunities to practice and experience these skills in real-world contexts. A growing body of literature, including works by Brand and Bellingham et al. [32,33], outlines core principles and key requirements for co-designing healthcare education and research to improve healthcare outcomes. Finally, increasing evidence underscores the importance of caregiver concerns and the integration of the worried sign of caregivers into early warning and response systems for critically ill patients, such as the Pediatric Early Warning System (PEWS) [21,34]. The parallels between caregiver concerns and caregiver’s TK, particularly in reading subtle signs, suggest that incorporating caregivers’ TK into clinical guidelines for physicians working with individuals with PIMD could be beneficial.

## 5. Conclusions

This study demonstrated that ID physicians understand caregivers’ TK in medical care for people with PIMD as vital for providing diagnostic cues and starting a joint diagnostic process. Effective communication about caregivers’ TK is essential for diagnosing and treating individuals with PIMD. However, the perceived gap between implicit caregivers’ TK and medical knowledge, difficulties to articulate TK, emotional entanglement, and conservatism are potential barriers to enabling such communication. To overcome these barriers, physicians should help to explicate caregivers’ TK and recognize it as complementary expertise that offers critical diagnostic cues for further medical investigation. Strategies that help to explicate caregivers’ TK include storytelling, demonstrations, joint observations, and measurements. Physicians’ sensitivity to emotions of fear and panic and feelings of uncertainty is paramount for fostering mutual trust and openness. Most importantly, communication about caregivers’ TK and integration with medical knowledge demands open communication, mutual trust, time and partnership between the physician and caregiver and may account for improved diagnosing and better medical care for people with PIMD.

## Figures and Tables

**Figure 1 healthcare-14-00025-f001:**
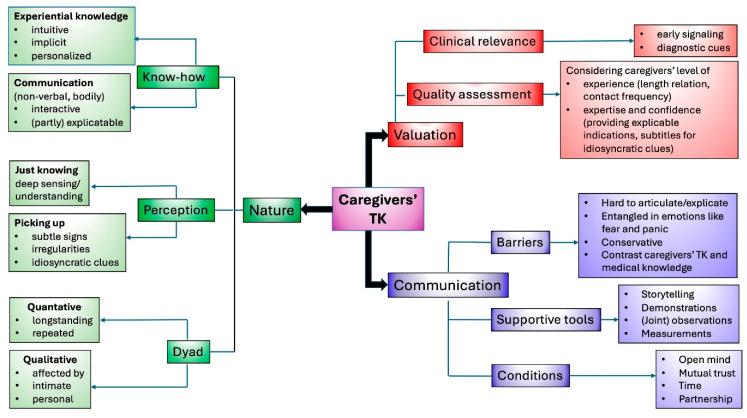
Thematic map of synthesis of findings on intellectual disability (ID) physicians’ perspectives on nature, valuation, and communication about caregivers’ tacit knowledge (TK) during consultations.

**Figure 2 healthcare-14-00025-f002:**
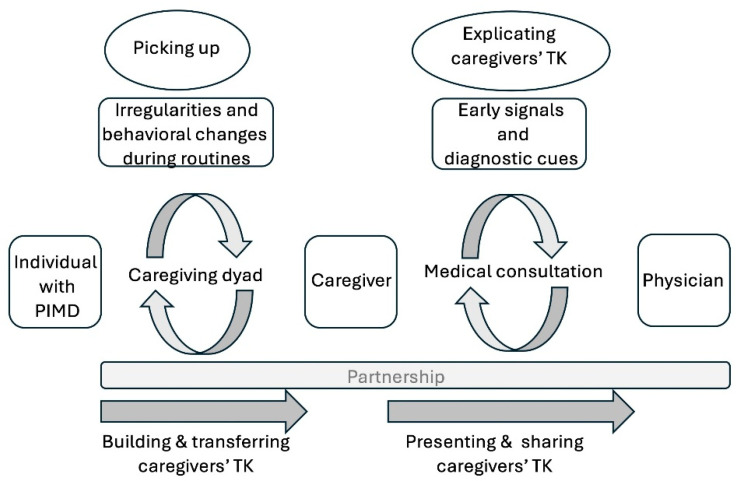
Thematic continuum how caregivers’ TK can be built and transferred in a caregiving dyad and how it can be presented and shared during medical consultation.

**Figure 3 healthcare-14-00025-f003:**
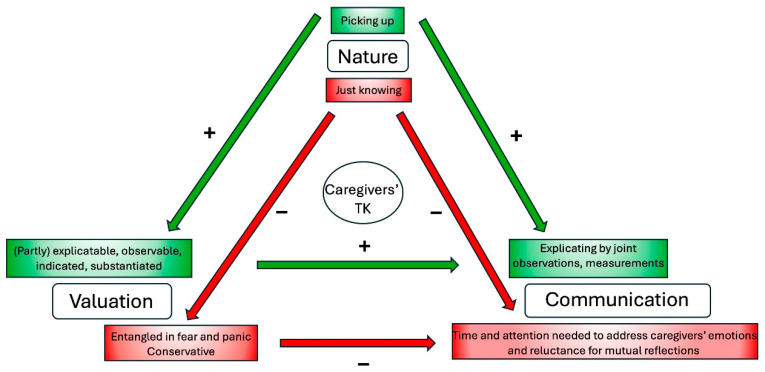
ID physicians’ perspectives on potential barriers (red) and subsequent facilitators (green) affecting communication during medical consultations in relation to the theme’s nature, valuation, and communication.

**Table 1 healthcare-14-00025-t001:** Characteristics of respondents.

Respondent (Gender)	Age (Range in Years)	Working Experience (Range in Years)	Work Setting (Residential/Out-Patient Clinic)
Physician 1 (female)	40–49	16–20	residential, out-patient clinic
Physician 2 (female)	50–59	16–20	residential, out-patient clinic
Physician 3 (female)	30–39	0–5	residential, out-patient clinic
Physician 4 (female)	50–59	6–10	residential clinic
Physician 5 (male)	30–39	0–5	residential clinic
Physician 6 (female)	40–49	0–5	residential, out-patient clinic
Physician 7 (female)	40–49	16–20	residential clinic
Physician 8 (male)	60–69	6–10	residential clinic
Physician 9 (female)	30–39	11–15	residential clinic
Physician 10 (female)	60–69	11–15	residential, out-patient clinic

Residential clinic (long-term care facility): for individuals with living arrangements; outpatient clinic: for individuals living with their parents/families.

## Data Availability

The data presented in this study are available on request from the corresponding author due to the qualitative nature of this study and privacy.

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
