# Peer review of "From Hidden Insights to Better Understanding: Physicians’ Perspectives on Caregivers’ Tacit Knowledge"

_healthcare, 2025, doi:10.3390/healthcare14010025_

Round 1
Reviewer 1 Report
Comments and Suggestions for Authors
The manuscript presents a structured and relevant qualitative study focused on the perceptions of physicians specialized in intellectual disability (ID physicians) regarding the tacit knowledge of caregivers of people with profound intellectual and multiple disabilities (PIMD). The topic is innovative, clinically and ethically significant, and addresses an underexplored area. The writing is clear, the argumentation coherent, and the conclusions consistent with the results. However, the manuscript requires greater methodological transparency and better alignment with the COREQ criteria, particularly regarding researcher description, reflexivity, ethical considerations, individual characterization of participants, and detail of the analytical process.
According to the COREQ checklist, the following areas for improvement are identified:
Domain 1. Research team and reflexivity
The study reports that the interviews were conducted by two researchers who are experts in intellectual disability, which lends professional credibility to the work. However, the COREQ guidelines recommend providing more detailed information on the researchers’ academic and training background, their experience with qualitative designs, and their potential influence on data generation. The manuscript does not report their training in qualitative research or previous experience. The researchers’ gender and epistemological stance are not specified. It is not clear whether there was any prior relationship with participants or what participants knew about the interviewers. There is no reflection on how their professional role (physicians interviewing colleagues) may have influenced responses, interview dynamics, or subsequent interpretation.
It is recommended to add a section titled “Research Team Reflexivity”, describing the authors’ positionality, the strategies used to enhance rigor and quality (reflexive diary, co-analysis, triangulation), and how their professional identity influenced the research.
Domain 2. Study design
The qualitative design is appropriately formulated, and the choice of semi-structured interviews is suitable for exploring professional perceptions. Nonetheless, several methodological components lack sufficient detail and should be expanded to fully meet COREQ requirements.
Aspects to improve include:
Theoretical orientation: The phenomenological design and theoretical framework of the study are not specified.
Sampling and participation: Inclusion and exclusion criteria are not detailed, nor is the number of physicians invited and those who declined to participate.
Sample description: Table 1 presents aggregated data but no individual-level participant information, which makes it difficult to link profiles to quotations. Qualitative studies should include a table with one row per participant (gender, age range, years of experience, type of practice).
Analytical procedures: The stages of thematic analysis are not described in terms of abstraction (meaning units, subthemes, themes), nor is it mentioned whether qualitative analysis software was used.
Field notes and validation: There is no mention of reflexive note-taking or participant review of transcripts.
Ethical considerations: Although ethical approval and informed consent are mentioned at the end of the manuscript, this information should be included within the Methods section. It is recommended to add a subsection titled Ethical considerations.
Domain 3. Analysis and findings
The thematic analysis is coherent, and the results are well structured into three main themes. The use of verbatim quotations is appropriate and enhances credibility.
Aspects to improve: The number of researchers involved in coding and whether cross-checking occurred are not specified. The analytical process (initial coding, theme development, final review) is not described. It is not indicated whether qualitative analysis software was used. There is no evidence of transcript or findings validation by participants.
General comments
Title: Too long and general.
Abstract: Add that the technique used was semi-structured interviews and that the analytical approach was interpretative. The objective should be clearly stated.
Methodology: Requires greater depth. The methods section should be more structured and include a description of the theoretical orientation and reflexivity aspects. Specifically, the analytical process (who coded, how themes were developed) must be detailed. Finally, ethical considerations should be included within the Methods section.
Results: Table 1 should be reformulated to include one row per participant with key demographic variables (gender, age, experience, work setting).
Discussion: Include limitations such as potential researcher bias, absence of triangulation, risk of social desirability bias, and limitations in international transferability. Add a brief statement on practical implications.
Incorporate a reflection on the researchers’ role and its potential influence on data collection and interpretation.
Author Response
|
Response to Reviewer 1 Comments
|
||
|
1. Summary |
|
|
|
We sincerely appreciate you taking the time to review this manuscript and for your valuable contributions to improving it. Thank you for your comments that the manuscript presents a structured and relevant qualitative study, that the topic is innovative, clinically and ethically significant, and addresses an underexplored area. Thank you for your valuable suggestions to improve methodological transparency and better alignment with the COREQ criteria, particularly regarding researcher description, reflexivity, ethical considerations, individual characterization of participants, and detail of the analytical process. Please, find our detailed responses below, along with the relevant revisions and corrections (via track changes) in the resubmitted files. |
||
|
2. Questions for General Evaluation |
Reviewer’s Evaluation |
Response and Revisions |
|
Does the introduction provide sufficient background and include all relevant references? |
Can be improved |
Please, see our response in the point-by-point response letter. |
|
Is the research design appropriate? |
Yes |
Please, see our response in the point-by-point response letter. |
|
Are the methods adequately described? |
Must be improved |
Please, see our response in the point-by-point response letter. |
|
Are the results clearly presented? |
Can be improved |
Please, see our response in the point-by-point response letter. |
|
Are the conclusions supported by the results? |
Can be improved |
Please, see our response in the point-by-point response letter. |
|
Are all figures and tables clear and well-presented? |
Can be improved |
Please, see our response in the point-by-point response letter. |
|
3. Point-by-point response to Comments and Suggestions for Authors |
||
|
Comments 1: Domain 1. Research team and reflexivity The study reports that the interviews were conducted by two researchers who are experts in intellectual disability, which lends professional credibility to the work. However, the COREQ guidelines recommend providing more detailed information on the researchers’ academic and training background, their experience with qualitative designs, and their potential influence on data generation. The manuscript does not report their training in qualitative research or previous experience. The researchers’ gender and epistemological stance are not specified. It is not clear whether there was any prior relationship with participants or what participants knew about the interviewers. There is no reflection on how their professional role (physicians interviewing colleagues) may have influenced responses, interview dynamics, or subsequent interpretation. It is recommended to add a section titled “Research Team Reflexivity”, describing the authors’ positionality, the strategies used to enhance rigor and quality (reflexive diary, co-analysis, triangulation), and how their professional identity influenced the research. |
||
|
Response 1: Thank you for pointing out that conducting the interviews by two researchers who are experts in intellectual disability, lends professional credibility to the work.
Our interdisciplinary research team consists of researchers from various fields (ethics, social sciences, and medical sciences). All members hold a PhD, with one serving as an emeritus professor. One of the ID physician researchers had formal training in qualitative research and has conducted several qualitative studies, while the other contributed as an early-career qualitative researcher with substantial clinical expertise. We added information about researchers’ academic and training backgrounds to the text.
We agree with the comment that we could have provided more explicit and complete information in the Methods section according to the COREQ guidelines. Therefore, we have made the necessary adjustments to address the items of the COREQ guidelines across the Methods section, except for the gender of the researchers as we found this less relevant. Please, see the added paragraph 2.2. Researchers’ positionality (page 3, lines 87-105) and revised sections in in paragraph 2.3 Recruitment (page 3, lines 109-125).
Reflection on how our professional roles (physicians interviewing colleagues) may have influenced responses, interview dynamics, or subsequent interpretation has also been pointed out in the Discussion section. |
||
|
Comments 2: Domain 2. Study design The qualitative design is appropriately formulated, and the choice of semi-structured interviews is suitable for exploring professional perceptions. Nonetheless, several methodological components lack sufficient detail and should be expanded to fully meet COREQ requirements. Aspects to improve include: Theoretical orientation: The phenomenological design and theoretical framework of the study are not specified. Sampling and participation: Inclusion and exclusion criteria are not detailed, nor is the number of physicians invited and those who declined to participate. Sample description: Table 1 presents aggregated data but no individual-level participant information, which makes it difficult to link profiles to quotations. Qualitative studies should include a table with one row per participant (gender, age range, years of experience, type of practice). Analytical procedures: The stages of thematic analysis are not described in terms of abstraction (meaning units, subthemes, themes), nor is it mentioned whether qualitative analysis software was used. Field notes and validation: There is no mention of reflexive note-taking or participant review of transcripts. Ethical considerations: Although ethical approval and informed consent are mentioned at the end of the manuscript, this information should be included within the Methods section. It is recommended to add a subsection titled Ethical considerations. |
||
|
Response 2: Following our response to comments 1, we concur again that we could have elaborated more on the aspects mentioned by the reviewer and have, accordingly, expanded these to fully meet COREQ requirements. Here, we will briefly point-by-point address these aspects.
Theoretical orientation: We discussed with the team whether we adopted a phenomenological approach or performed more like a multiple case study, but we concluded that it most appropriate to consider this a qualitative study with semi-structured interviews.
Sampling and participation: Inclusion and exclusion criteria have been described more explicit, including the number of physicians invited and those who declined to participate (page …, lines ….).
Sample description: We changed Table 1 to present individual-level participant information with one row per participant (gender, age range, years of experience, work setting) (page 5).
Analytical procedures: In 2.5 Data analysis (page 4, lines 157-171), we described the stages of thematic analysis and the qualitative analysis software (MaxQDA) we used.
Field notes and validation: No reflexive note-taking or participant review of transcripts was conducted. Reflexivity was ensured through regular reflective and methodological discussions with our interdisciplinary research team throughout the study (page 3, lines 94-95) as well as by reviewing and discussing our findings and interpretations in meetings and conferences with parents, professional caregivers, ID physicians, developmental pediatricians, and other professionals involved in the care of individuals with PIMD (page 4, 165-169).
Ethical considerations: We included 2.6 Research ethics within the Methods section (page 4, lines 174-183). |
||
|
Comments 3: Domain 3. Analysis and findings The thematic analysis is coherent, and the results are well structured into three main themes. The use of verbatim quotations is appropriate and enhances credibility. Aspects to improve: The number of researchers involved in coding and whether cross-checking occurred are not specified. The analytical process (initial coding, theme development, final review) is not described. It is not indicated whether qualitative analysis software was used. There is no evidence of transcript or findings validation by participants. |
||
|
Response 3: We thank the reviewer again for addressing this point, and have the analytical process and qualitative analysis software described in 2.5 Data analysis (page 4, line 158) |
||
|
Comments 4: Title: Too long and general.
Abstract: Add that the technique used was semi-structured interviews and that the analytical approach was interpretative. The objective should be clearly stated.
Methodology: Requires greater depth. The methods section should be more structured and include a description of the theoretical orientation and reflexivity aspects. Specifically, the analytical process (who coded, how themes were developed) must be detailed. Finally, ethical considerations should be included within the Methods section.
Results: Table 1 should be reformulated to include one row per participant with key demographic variables (gender, age, experience, work setting).
Discussion: Include limitations such as potential researcher bias, absence of triangulation, risk of social desirability bias, and limitations in international transferability. Add a brief statement on practical implications.
Incorporate a reflection on the researchers’ role and its potential influence on data collection and interpretation. |
||
|
Response 4: We thank the reviewer for the suggestions and will address these point-by-point.
Title: Too long and general. We changed the title into: From Hidden Insights to Better Understanding; Physicians’ perspectives on Caregivers’ Tacit Knowledge
Abstract: The abstract describes the objective, the use of semi-structured interviews. We added that the analytical approach was interpretative (page 2, line 43).
Methodology: In the methods section we added paragraph 2.2 Researchers’ positionality (page 3, lines 87-105) and paragraph 2.6 Research ethics (page 4, lines 174-183) and improved structure and content, specifically regarding the analytical process (2.5 Data analysis, page 4, lines 156-171)
Results: We reformulated Table 1 including one row per participant with key demographic variables (gender, age, experience, work setting) (page 5).
Discussion: We incorporated a reflection on potential researcher bias, absence of triangulation, risk of social desirability bias, and limitations in international transferability in paragraph 4.2. Strengths and limitations (page 10-11, lines 421-438/450). We elaborated on practical implications in 4.7 Knowledge mobilization (page 14-15, lines 586-623) |
||
|
4. Response to Comments on the Quality of English Language |
||
|
No comments |
||
|
|
||
|
5. Additional clarifications |
||
|
None |
||
Reviewer 2 Report
Comments and Suggestions for Authors
1.Suggestion to strengthen theory and literature
The manuscript offers valuable insights and presents rich qualitative data; however, the theoretical grounding of tacit knowledge could be further strengthened. I kindly suggest expanding the literature review to include foundational work on tacit knowledge and professional expertise (e.g., Polanyi, Benner, Schön) as well as relevant frameworks on knowledge communication or co-production. Incorporating these perspectives would help clarify the conceptual basis of TK and enhance the broader contribution of the study.
2.Introduction can be more concise
Some background paragraphs repeat similar ideas (non-verbal cues, reliance on caregivers). Consider tightening the text to increase clarity.
3.Terminology consistency
Ensure consistent use of terms:
“caregivers” vs. “parents and professional caregivers,”
“tacit knowledge” vs. “experiential knowledge.”
Unify terminology to avoid conceptual overlap.
4.Add practical implications
Although implications are discussed, adding concrete examples of:
how physicians can systematically incorporate TK, how caregiver training or communication tools might support this, would increase usefulness for practitioners.
5.Check minor language/grammar issues
Overall English is strong; only occasional long sentences could be shortened.
Author Response
|
Response to Reviewer 2 Comments
|
||
|
1. Summary |
|
|
|
We sincerely appreciate you taking the time to review this manuscript and for your valuable contributions to improving it. Please, find our detailed responses below, along with the relevant revisions and corrections (via track changes) in the resubmitted files. |
||
|
2. Questions for General Evaluation |
Reviewer’s Evaluation |
Response and Revisions |
|
Does the introduction provide sufficient background and include all relevant references? |
Can be improved |
Please, see our response in the point-by-point response letter. |
|
Is the research design appropriate? |
Yes |
Please, see our response in the point-by-point response letter. |
|
Are the methods adequately described? |
Yes |
Please, see our response in the point-by-point response letter. |
|
Are the results clearly presented? |
Can be improved |
Please, see our response in the point-by-point response letter. |
|
Are the conclusions supported by the results? |
Can be improved |
Please, see our response in the point-by-point response letter. |
|
Are all figures and tables clear and well-presented? |
Yes |
Please, see our response in the point-by-point response letter. |
|
3. Point-by-point response to Comments and Suggestions for Authors |
||
|
Comments 1: Suggestion to strengthen theory and literature The manuscript offers valuable insights and presents rich qualitative data; however, the theoretical grounding of tacit knowledge could be further strengthened. I kindly suggest expanding the literature review to include foundational work on tacit knowledge and professional expertise (e.g., Polanyi, Benner, Schön) as well as relevant frameworks on knowledge communication or co-production. Incorporating these perspectives would help clarify the conceptual basis of TK and enhance the broader contribution of the study. |
||
|
Response 1: Thank you for pointing out that our manuscript offers valuable insights and presents rich qualitative data. We agree with your comment that the theoretical grounding of tacit knowledge could be further strengthened. Therefore, we have expanded the literature review accordingly, to include foundational work by Polanyi, Smith, Schön, Benner and Pope on tacit knowledge and professional expertise (paragraph 4.3. The emergence of caregivers’ TK in PIMD studies, page 11, lines 456-466). For the same cause of theoretical grounding we added also works on knowledge communication and co-designing by Brand and Bellingham (paragraph 4.7. Knowledge mobilization, page 15, lines 614-617). We hope that incorporating these theoretical backgrounds would help to clarify the conceptual basis of TK and enhance the broader contribution of the study towards partnership and co-designing healthcare education and research to improve healthcare outcomes. |
||
|
Comments 2: Introduction can be more concise Some background paragraphs repeat similar ideas (non-verbal cues, reliance on caregivers). Consider tightening the text to increase clarity. |
||
|
Response 2: Thank you for your suggestion. In the introduction, we removed non-verbal cues (page 2, line 62) to avoid similarity with non-verbal expressions (page 2, lines 65).
Furthermore, in the background section of the abstract, we removed the sentence that ‘individuals rely on their caregivers’ expertise ……’ (page 1, lines 35-37) and tightened the text to increase clarity. |
||
|
Comments 3: Terminology consistency Ensure consistent use of terms: “caregivers” vs. “parents and professional caregivers,” “tacit knowledge” vs. “experiential knowledge.” Unify terminology to avoid conceptual overlap. |
||
|
Response 3: Thank you for pointing out these nuances. With the broader term “caregivers”, as we use in caregivers’ TK, we refer to a broader group including both parents and professional caregivers. We checked the manuscript on consistent use of the term parents when we meant parents and the term professional caregivers when we meant professional caregivers working in residential or daycare facilities. We refer to tacit knowledge, when we specifically mean experiential knowledge that is particularly hard to articulate. We checked the manuscript on consistent use of the terms experiential knowledge and tacit knowledge. |
||
|
Comments 4: Add practical implications Although implications are discussed, adding concrete examples of: how physicians can systematically incorporate TK, how caregiver training or communication tools might support this, would increase usefulness for practitioners. |
||
|
Response 4: Thank you for your suggestions to make the implications more useful for practitioners. In the Discussion section we point out that trust and partnership (paragraph 4.5 Trust and partnership, page 12, lines 515-537) and effective communication (paragraph 4.6 Effective communications, page 13, lines 538-555) are preconditions for incorporating caregivers’ TK during medical consultations. In paragraph 4.6.2. Facilitators (page 14, lines 565-578), we refer to both communications competencies (e.g. sensitive and responsive to caregivers’ emotions to increase trust and an open dialogue) and tools (e.g. like storytelling, demonstrations and joint observations) that physicians may acquire and engage to be able to incorporate caregivers’ TK. In paragraph 4.7 Knowledge mobilization we added that communication and partnership can be enhanced through educational activities including conversations, storytelling, demonstrations, joint observations, speed dating sessions with EBE’s, and home visits, which provide opportunities to practice and experience these skills in real-world contexts (page 15, lines 611-614). Finally, we suggest that incorporating caregivers' TK into clinical guidelines for physicians working with individuals with PIMD could be beneficial, given the parallels between caregiver concerns and caregiver’s TK, particularly in reading subtle signs (page 15, lines 617-623). |
||
|
4. Response to Comments on the Quality of English Language |
||
|
Point 1: Check minor language/grammar issues Overall English is strong; only occasional long sentences could be shortened. |
||
|
Response 1: We checked language and grammar, made minor corrections and shortened some long sentences. |
||
|
5. Additional clarifications |
||
|
None |
||